# Anti-Biofilm Activity of Phenyllactic Acid against Clinical Isolates of Fluconazole-Resistant *Candida albicans*

**DOI:** 10.3390/jof9030355

**Published:** 2023-03-15

**Authors:** Angela Maione, Marianna Imparato, Annalisa Buonanno, Federica Carraturo, Antonetta Schettino, Maria Teresa Schettino, Marilena Galdiero, Elisabetta de Alteriis, Marco Guida, Emilia Galdiero

**Affiliations:** 1Department of Biology, University of Naples ‘Federico II’, Via Cinthia, 80126 Naples, Italy; 2Department of Experimental Medicine, University of Campania “Luigi Vanvitelli”, 80138 Naples, Italy; 3Department of Ginecologia ed Ostetricia, AOU I Policlinico Università della Campania Luigi Vanvitelli, 80138 Naples, Italy

**Keywords:** *Candida albicans*, fluconazole resistance, phenyllactic acid, biofilm, *Galleria mellonella*

## Abstract

Commonly found colonizing the human microbiota, *Candida albicans* is a microorganism known for its ability to cause infections, mainly in the vulvovaginal region, and is responsible for 85% to 90% of vulvovaginal candidiasis (VVC) cases. The development of drug resistance in *C. albicans* isolates after long-term therapy with fluconazole is an important complication to solve and new therapeutic strategies are required to target this organism and its pathogenicity. In the present study, phenyllactic acid (PLA) an important broad-spectrum antimicrobial compound was investigated for its antifungal and antivirulence activities against clinical isolates of *C. albicans*. Previously characterized strains of *C. albicans* isolates from women with VVC and *C. albicans* ATCC90028 were used to evaluate the antimicrobial and time dependent killing assay activity of PLA showing a MIC 7.5 mg mL^−1^ and a complete reduction of viable *Candida* cells detected by killing kinetics after 4 h of treatment with PLA. Additionally, PLA significantly reduced the biomass and the metabolic activity of *C. albicans* biofilms and impaired biofilm formation also with changes in *ERG11*, *ALS3*, and *HWP1* genes expression as detected by qPCR. PLA eradicated pre-formed biofilms as showed also with confocal laser scanning microscopy (CLSM) observations. Furthermore, the compound prolonged the survival rate of *Galleria mellonella* infected by *C. albicans* isolates. These results indicate that PLA is a promising candidate as novel and safe antifungal agents for the treatment of vulvovaginal candidiasis.

## 1. Introduction

Vulvovaginal candidiasis (VVC) is one of the most common clinical conditions caused by *Candida* species [1]. Epidemiological studies report that about 75% of women have had at least one episode during their lifetime, and of these women 10%, wearing mobile prosthetic replacements such as catheters and intrauterine devices (IUDs), have recurrent infection (RVVC) affecting their quality of life [2]. Risk factors associated to this pathology could be decrease in immunity, overuse of antibiotics, pregnancy, or uncontrolled diabetes mellitus causing symptomatic infection and, as a consequence, plenty mucosal inflammation [3]. The majorities of VVC are caused by the abnormal growth of *Candida* spp., normally present in the vaginal flora, with the transition of the fungus from ovoid yeast-like cells to elongated hyphae and the formation of biofilm. *Candida albicans*, which is part of normal vaginal microflora and becomes a robust opportunistic fungal pathogen under various conditions, has been considered the major causes of vulvovaginitis, the medical impact of which also depending on the strain ability to form biofilms [4]. Regarding treatment, intravaginal and oral azole drugs are currently prescribed for the treatment of VVC and Fluconazole, which is the drug of choice led to the acquisition of inherent resistance, in about 30% of vaginal positive cultures, consequently resulting in failure of therapeutic treatment of the infection [5,6]. Biofilms are communities of microorganisms embedded in an exopolysaccharide matrix that attach to surfaces. Within biofilms, involved in almost 80% of human infections, microorganisms are less susceptible to drugs, so as to favor the persistence of the infection [7].

Therefore, in recent years, much attention has been paid to natural innovative substances and alternative technological solutions, such as the green compound Phenyllactic acid [8]. Phenyllactic acid (PLA) is an organic acid with broad-spectrum antimicrobial activity, used as preservative in the food industry with capacity to prolong food shelf life [9,10]. PLA is also widely used in many other fields, such as medicine, feed, cosmetics, and gel materials, due to its non-toxicity to humans [11,12]. Recently, a study identified the microbial metabolite phenyllactic acid as a candidate inhibitor of tyrosinase and melanogenesis [13]. Although the antibacterial activity of PLA against several foodborne pathogenic bacteria has been well documented [14], as well as emerging reports on its anti-biofilm activity against *Enterobacter cloacae* [15], *Pseudomonas aeruginosa* [16], *Listeria monocytogenes* [17], and *Enterococcus faecalis* [18], to the best of our knowledge, the antifungal and antibiofilm activity of PLA against *C. albicans* has not yet been fully explored. Furthermore, no reports are available concerning the application of PLA for the control of VVC. Therefore, in this study we carried out investigations aimed to (i) evaluate the antifungal activity of PLA against clinical isolates of fluconazole-resistant *C. albicans* in both planktonic and biofilm states, (ii) explore the potential molecular action underlying the antibiofilm activity of PLA, and (iii) assess the antifungal performance of PLA in vivo using a *Galleria mellonella* larvae model.

## 2. Materials and Methods

### 2.1. Ethics Statement

A written informed consent form was filled out and obtained by each patient. All clinical and demographic data were kept confidential and data in this study were processed anonymously. The research conforms to the ethical guidelines of the Helsinki Declaration. Ethical approval n. 0022538/i dated 20 July 2022 was granted by the University of Campania Luigi Vanvitelli.

### 2.2. Sample Collection

Women with suspected clinical evidence of VVC were examined by the specialist gynecologist from the gynecology clinic at the University of Naples, Italy during September–December 2022 and sampled using two vaginal secretion swabs from each patient. Pregnant women and patients with diabetes, malignant tumors, immunodeficiency, or treated with antibiotics for a long period were excluded from the study. Fifty patients suspected of VVC were identified and their identifying information and patient demographic characteristics, including age, previous vaginitis, clinical symptoms, diabetes, contraceptive used, and antibiotic used, were kept confidential. All participants signed a written consent form.

### 2.3. Identification of Candida spp.

The clinical samples for fungal culture were inoculated into Sabouraud Dextrose agar (SDA, Merck, Germany) supplemented with 0.5% (*w*/*v*) chloramphenicol (Sigma Aldrich, Saint Louis, MO, USA) and incubated at 37 °C for 24/48 h. The *Candida* spp. isolates were identified using the phenotyping method and confirmed by PCR. A presumptive identification of *Candida* species based on the morphology and colors of the colonies was performed by culturing on CHROM agar (Merck, Germany) at 37 °C for 24–48 h. To confirm the detection of *Candida albicans* species, the fungal strains identification was performed by molecular identification methods using PCR technique as reported elsewhere [19,20] and compared with reference strains using nucleotide blast at GenBank (https://www.ncbi.nlm.nih.gov/genbank/ accessed on 9 January 2023). Then, all strains were kept in 10% glycerol at −80 °C, sub-cultured in Sabouraud dextrose agar (SDA, Merck, Germany) and cultured in Tryptone Soya Broth (TSB, OXOID, Basingstoke, UK), supplemented with 1% *w*/*v* glucose for 24 h at 37 °C in order to recover yeast cells, then washed twice using phosphate-buffered saline (PBS) (OXOID, Basingstoke, UK) and standardized to 10^6^ cells mL^−1^. In addition, the reference strain of *C. albicans* ATCC90028 was cultured in the same conditions and used throughout the study.

### 2.4. Antifungal Susceptibility Testing

The susceptibility of *Candida* clinical isolates and *C. albicans* ATCC90028 to three antifungal drugs was carried out by the disc diffusion method as described by Clinical and Laboratory Standards Institute [21]. A zone diameter of ≥19 mm was considered sensitive, 15 to 18 mm was considered intermediate, and a diameter ≤14 mm was considered resistant. The tested antifungal drugs included amphotericin B (AMB) (1 μg), fluconazole (FLC) (25 μg), and caspofungin (CSF) (5 μg), supplied by Sigma-Aldrich (Sigma-Aldrich Saint Louis, MO, USA).

The minimum inhibitory concentration (MIC) and the minimum fungicidal concentration (MFC) of PLA were detected only for *C. albicans* resistant to fluconazole into 96-well plates according to CLSI guidelines (M27-S4) [22]. PLA (Sigma-Aldrich, Saint Louis, MO, USA) was dissolved in DMSO 5% *v*/*v* to prepare the 100 mg mL^−1^ stock solution. Briefly, fungal cultures were incubated in RPMI medium at 37 °C for 24 h together with PLA ranging from 1 to 25 mg mL^−1^ and growth was determined at 590 nm wavelength with a microplate reader (SYNERGYH4 BioTek Agilent, Technologies, Tigan St, Winooski VT05404 USA). The MIC was defined as the lowest concentration of the sample capable of inhibiting fungal growth. In addition, for the determination of the MFC, an aliquot of 5 μL from each well with no evident fungal growth was transferred to SDA plates and incubated at 37 °C, for 48 h. The MFC was defined as the smallest concentration that inhibited growth on the agar plates. The MFC/MIC ratio was calculated to determine whether the substance had fungistatic (MFC/MIC ≥ 4) or fungicidal (MFC/MIC < 4) activity [23]. All the assays were performed in triplicates and repeated at least three different times for reproducibility.

### 2.5. Time-Kill Kinetics Assay of the PLA

The time-kill kinetics of PLA was carried out, similar to our previous protocol [24] designed by Flamm et al. [25], with slight modification. Briefly, each four-fluconazole resistant *C. albicans* clinical strain (C7, C14, C17, and C19) as well as the ATCC90028 strain at a concentration of 1·10^6^ CFU·mL^−1^ were inoculated in the culture medium containing PLA at concentrations equivalent to ½ MIC, MIC and 2xMIC and incubated at 37 °C for 24 h. Aliquots of 1 mL, properly diluted, were pipetted into Petri dishes at intervals of 1 h for the first 8 h and after 24 h, for each tested organism, and incubated at 37 °C for 24 h. The mean of the colony-forming unit (CFU) count was used to determine the viable cells. The experiment was carried out in triplicate for each strain.

### 2.6. Biofilm Formation Studies

The biofilm-forming potentials of the clinical fluconazole-resistant *C. albicans* cultures and *C. albicans* reference strain was investigated by our previous protocol [24]. Briefly, to form biofilms, 96-well sterile flat-bottomed microplate was seeded with 10^6^ cells mL^−1^ inoculum of test organisms (100 μL per well), followed by incubation for 24/48 h at 37 °C. Total biofilm mass was quantified using the crystal violet (CV) staining methodology. Briefly, the plates were washed three times with PBS, and the biofilm was fixed at 37 °C for 1 h. In total, 200 µL of CV (0.2% *v*/*v*) was added to well, and after 15 min, the excess was washed with PBS. Biofilm was resuspended by adding 300 µL of 30% (*v*/*v*) acetic acid. The absorbance was quantified at 570 nm using a microtiter plate reader, as described previously [26]. OD cut-off (ODc) was calculated to classify the ability of yeast to form biofilm using the following formula: OD_570_ mean of negative control plus three time the standard deviation (SD). The classification was: negative (OD ≤ ODc), weak (ODc ≤ OD ≤ 2×ODc), moderate (2×ODc < OD), or strong (OD ≥ 4×ODc). Biofilms vital biomass was quantified by using the tetrazolium 2,3-bis (2-methoxy-4-nitro-5 sulfophenyl)-5-[(phenylamine) carbonyl]- 2H-hydroxide reduction assay (XTT) (Sigma-Aldrich, St. Louis, MO, USA) according to the manufacturer’s instructions. The absorbance of resulting solution was measured at 450 nm using a microtiter plate reader [27]. Each of the procedures was performed in triplicate.

### 2.7. Biofilm Treatment

To evaluate the PLA activity on the inhibition of biofilm formation, strain suspensions were added into a 96-well microplate wells in the presence of different concentrations of the compound ranging from 0.6 to 5 mg mL^−1^, and incubated at 37 °C for 24 h. To assess the ability of compound to disrupt a mature biofilm, after 24 h of incubation at 37 °C, the cells were washed twice with PBS to remove planktonic yeast, and PLA ranging from 0.6 mg mL^−1^ to 5 mg mL^−1^ was added. The plates were incubated for another 24 h at 37 °C. Following both treatments, all biofilms were rinsed once with PBS to remove non-adhered cells, and a crystal violet (CV) staining and XTT reduction assay were employed, respectively, to quantify the total biofilm biomass and metabolic activity of biofilm cells, as above described. Biofilm reduction was calculated using the following equation: % Biofilm reduction: (Abs_control_ − Abs_PLA_)/(Abs_control_) × 100 and % Viability: (Abs_PLA_)/(Abs_control_) × 100, where Abs_control_: absorbance of the control and Abs_PLA_: absorbance of the sample treated with PLA.

### 2.8. Confocal Laser Scanner Microscope Observations of a Selected Biofilm

A mature biofilm of the *C. albicans* ATCCC 90028 was incubated for 24 h in the presence of PLA 2.5 mg L^−1^ to assess its anti-biofilm (eradication) activity evaluated by Confocal Laser Scanning Microscopy (CLSM) [27]. Aiming at this, untreated and PLA-treated biofilms formed on the bottom of a multi-well plate for high resolution microscopy (NuncTM Lab-Tek^®^ 8-well Chamber Slides (n◦17744; Thermo Scientific, Ottawa, ON, Canada)) were washed with PBS and stained with Calcofluor 0.25 mg mL^−1^ (Sigma-Aldrich, Saint Louis, MO, USA) for 30 min at room temperature, in the dark. Observations were carried out using an inverted confocal laser-scanning microscope (Zeiss LSM700) by capturing images at 10×.

### 2.9. Quantitative Real-Time PCR (qRT-PCR)

The potential mechanisms of PLA to inhibit biofilms was investigated trough gene expression analysis of some genes involved in biofilm formation. In turn, the four clinical isolates of *C. albicans* and reference strain were evaluated for the expression of *ERG11, ALS3*, and *HWP1* genes using qRT-PCR. The primers used to amplify and identify expression of *ERG11*, *ALS3*, *HWP1*, and *Act1* as an internal reference gene are reported in Table 1. RNA extraction and real-time PCRs were performed as described previously [24]. Briefly, fungal cells of biofilm grown together with PLA (2.5 mg mL^−1^) at 37 °C for 24 h were scraped and washed in PBS. Total RNA was extracted using Direct-zolTM RNA Miniprep Plus Kit (ZYMO RESEARCH, Irvine, CA, USA) according to the manufacturer’s protocol. The cDNA template was obtained from the total RNA using cDNA synthesis kit QuantiTect Reverse Transcription Kit (Qiagen, Valencia, CA, USA), following the manufacturer’s protocol. The QuantiTect SYBR Green PCR Kit (Qiagen, Valencia, CA, USA) was used to perform the Real-Time PCR.

Briefly, 25 µL of SYBR Green was added with to 100 ng of cDNA, 1 µM of each primer, and 12.5 µL of QuantiFast SYBR Green PCR Master Mix (2×). PCR cycling was performed as described in previous study [27]. All steps were done in a Rotor-Gene Q cycler (Qiagen, Valencia, CA, USA). The expression levels of genes were evaluated using the 2^−∆∆CT^ method, where Ct was the average threshold cycle number from three independent experiments. Data were presented as the fold change in gene expression normalized to the *Act1* gene as internal control [28,29].

**Table 1 jof-09-00355-t001:** Primer sequences used in this study.

Gene Name	Acronym	Primer Name	Sequence (5′→3′)	Tm (°C)	Amplicon Length (bp)	Reference
*Actin*	*actin*	C.albicans_actin_F	AGCCCAATCCAAAAGAGGTATT	62	153	[30]
		C.albicans_actin_R	GCTTCGGTCAACAAAACTGG	63		
*Ergosterol biosynthesis enzyme*	*ERG11*	C.albicans_ERG11_F	ATTGTTGAAACTGTCATTG	62	185	[31]
		C.albicans_ERG11_R	CCCCTAATAATATACTGATCTG	63		
*Agglutinin like-sequence 3*	*ALS3*	C.albicans_ALS3_F	CTAATGCTGCTACGTATAATT	56	201	[30]
		C.albicans_ALS3_R	CCTGAAATTGACATGTAGCA	58		
*Hyphal wall protein 1*	*HWP1*	C.albicans_HWP1_F	CAGCCACTGAAACACCAACT	63	135	[30]
		C.albicans_HWP1_R	CAGAAGTAACAACAACAACACCAG	63		

### 2.10. Galleria Mellonella Survival Assay

To evaluate the PLA toxicity and its antimicrobial effect in vivo, a *G. mellonella* survival test was performed as described previously [30,32]. Briefly, to test PLA toxicity, different concentrations (1.25, 2.5, 5, 10, and 20 mg L^−1^) of compound were injected into the last left proleg of larvae and their survival was monitored every 24 h for 3 days. One concentration of PLA (2.5 mg mL^−1^) was selected to conduct experiments of prevention and therapeutic treatment of candida infection in vivo. The standard fungal suspensions (10^6^ CFU mL ^–1^) were prepared in PBS from the overnight cultures. Cells were directly injected with a 10 mL Hamilton syringe in the last left proleg. PLA at a concentration of 2.5 mg mL^−1^ was administered to larvae 2 h before infection (prophylactic treatment) or 2 h after infection (therapeutic treatment), who were incubated at 37 °C for 72 h, and the survival was monitored every 24 h. The larvae were considered to have died when they did not respond to physical stimulation.

Different groups, each containing 20 randomly selected larvae, were included for the assay: (a) not injected, to evaluate general viability, (b) injected with PBS buffer only, and (c) injected with each of the strain (ATTC 90028, C7, C14, C17, and C19); thus, we had untreated, pre-treated, and post-treated groups. Experiments were repeated twice.

### 2.11. Statistical Analysis

GraphPad Prism Software (version 8.02 for Windows, GraphPad Software, La Jolla, CA, USA, www.graphpad.com, accessed on 23 January 2023) was used for statistical analysis. Data are represented as mean ± standard deviation (SD). In the *G. mellonella* model, the survival curves were plotted using Kaplan–Meier method. One-way ANOVA followed by Dunnett’s post hoc test and two-way ANOVA followed by Tukey’s test were used to compare the differences among the groups. Differences were considered significant when the *p*-value was <0.05.

## 3. Results

### 3.1. Microorganisms: Selection and Identification

The collection period was from September 2022 to December 2022. Samples from 50 women were obtained, of which 62% (31/50) showed positive culture for yeasts. Molecular analysis, see Table 2, showed that the most common species isolated was *C. albicans* at 71% (22/31), followed by *C. dubliniensis* at 10% (3/31), *C. glabrata* at 6% (2/31), *C. haemulonii* at 6% (2/31), *C. tropicalis* at 3% (1/31), and *C. orthopsilosis* at 3% (1/31). The 22 *C. albicans* isolated from vaginal content were tested, by disc diffusion method, against the drugs: AMB, CSF, and FLC results are described in Table 3. *C. albicans* (ATCC90028) was included for quality control tests. According to the susceptibility pattern, all of *C. albicans* isolates were susceptible to AMB (99.8%), as well as CS (90%) and FLC (77%). Our results showed that for CSF, only the isolate C14 had an alone of 15 mm diameter so as considered intermediate and C19 that was resistant. On the other hand, fluconazole showed susceptibility in most of the strain isolated, except for C7, C14, C17, and C19, which were clearly resistant, and C24, which was considered intermediate. Two *C. albicans* isolates were resistant to both FLC and CSF, i.e., C14 and C19, even if strain C14 was considered intermediate because it was resistant to CSF in a dose-dependent manner.

### 3.2. Biofilm Formation

We detected the biofilm formation ability of the four clinical strains which had resulted FLC resistant and the ATCC90028, as reported in Figure 1 (panel A,B). The amount of biofilm developed, at 24 and 48 h, was determined by quantifying the total biomass of biofilm cells with crystal violet and by measuring the metabolic activity of biofilm cells using a XTT reduction assay. The result obtained showed that each *C. albicans* strains was able to form considerable amount of biofilm. As reported in Figure 1 (panel A), all strains were able to form a strong biofilm at 24 and 48 h, except for the C14 isolate. The maximum amount of total biofilm biomass was recovered for the reference strain and C19 at both the observation times. This was in line with the XTT data (Figure 1, panel B), where the maximum amount of vital biofilm biomass was recovered in both C19 and the reference strain.

### 3.3. Susceptibility of C. albicans Planktonic Cells to PLA

Considering these preliminary results, we evaluated the antifungal efficacy of PLA towards FLC-resistant and biofilm-forming clinical samples. Table 4 shows individual MIC and MFC values to PLA for each of the *C. albicans* strains. In the broth dilution assay, *Candida* cells in the inoculums were reduced in a dose-dependent manner by the presence of PLA. The MIC value was 7.5 mg mL^−1^ for all the four clinical strains as well as for *C. albicans* reference strain showing an efficient antimicrobial activity. The MFC was similar to MIC for C7, C14, and the reference strain and 10 mg mL^−1^ for C17 and C19. According to the parameters stated previously (10.2147/IDR.S164262), the MFC/MIC ratio demonstrated a fungicidal effect of the PLA against all the species tested.

Time-kill curves were performed for both the FLC-resistant *C. albicans* and the reference strains, using different concentrations of PLA ranging from 0.5 to 2 × MIC. The results obtained for the time-kill curves are summarized in Figure 2 for each strain. The effect of PLA on the yeast growth curves and the rapid decrease in fungal vitality are clearly visible, with differences and a strong positive correlation between PLA concentrations and yeast growth reduction for all the *C. albicans* strains. In particular, PLA at MIC concentration exerted a 2-log-decrease for the ATCCstrain, C17 and C19 isolates, and a 4-log decrease for the C7 isolate after 4 h and a 6-log reduction until complete killing for the C14 after only 2 h.

### 3.4. Effect of PLA in Biofilm Inhibition and Eradication of C. albicans

Based on the susceptibility of planktonic *C. albicans* cells to PLA, we evaluated the effect of this drug at sub-MIC concentrations on biofilms formation and on pre-formed biofilms. The effect of PLA different concentrations ranging from 0.6 to 5 mg mL^−1^ on *C. albicans* biofilm formation was studied. Biofilm formed in the absence of the compound was used as negative control. All the *C. albicans* strains showed reduced biofilm formation in the presence of different concentrations of PLA (Figure 3 panel A). The biofilm inhibition potentials varied from 10–30% inhibition for the lowest concentration tested to 50–65% for the highest concentration tested for all clinical isolates. The reference strain showed 95% inhibition at the highest concentration tested, suggesting that during biofilm formation, the virulence in clinical isolates was less involved compared to reference strain. The effect of PLA on the viability was detected by XTT and reported in Figure 3 (panel B). The viability of all candida cells had a significant reduction with increasing PLA concentration compared to the control group (100% viability). We observed that at 2.5 mg mL^−1^, the viability percentage was of 30–35%. In Figure 4 (panel A–B) is reported the PLA biofilm reduction ability on mature biofilms. A biofilm biomass eradication percentage between 40 and 80% was obtained after 24 h of exposure to PLA, confirmed by a significant reduction in viability. The lowest PLA activity was reported for the C19 isolates, resistant to both FLC and CSF. These results showed that the biofilm inhibition and eradication effects of PLA were strain dependent.

### 3.5. Biofilm Visualization by CLSM

To support our results, CLSM was used to visualize biofilm structure before and after the treatment with PLA. In particular, Figure 5 (panel A,B) confirmed that the biofilm was significantly affected by PLA, which was able to eradicate the pre-formed biofilm, showing an evident biomass reduction compared to the untreated biofilm. In Figure 5 (panel A), a very compact biofilm structure was observed in the untreated sample, while the structural destruction in most of the biofilm was evident following PLA treatment.

### 3.6. Effect of PLA on ERG11, ALS3, and HWP1 Gene Expression

To elucidate the potential molecular mechanism of PLA on *C. albicans* biofilms (reference strain, C7, C14, C17, and C19), we examined the changes in the expression levels of biofilm-related genes (*ERG11*, *ALS3*, and *HWP1*) using qRT-PCR. The expression pattern of the three selected genes is shown in Figure 6. The compound significantly down-regulated *ALS3* in all strains and *ERG11* in reference strain, C17, and C19 after treatment with 2.5 mg mL^−1^ compared to control. *HWP1* was unaffected by the treatment and down-regulated only in C17. Taken together, the qRT-PCR results indicate that treatment with compound may affect biofilm formation by reducing the levels of adhesins without interesting the transition from yeast to hyphae.

### 3.7. Toxicity Assessment and Evaluation of PLA Treatment of Experimental Infection by C. albicans in Galleria mellonella

Predicting in vivo toxicity is of fundamental importance, so it has been evaluating the toxicity of PLA through *G. mellonella* larvae an alternative in vivo model that gain increasingly space due to low cost, low biological risk, and the similarity of the immune system of these larvae to mammals innate immune response, which makes their use even more desirable in predicting toxicity in humans. As can be seen in Figure 7, PLA at low doses (1.25 and 2.5 mg mL^−1^) did not cause death or toxicity signs in the larvae, suggesting that their use can be considered safe. On the contrary, at the highest concentrations tested, i.e., 20 mg mL^−1^ and 10 mg mL^−1^, the observed survival decreased by 100% in 24 h for the concentration of 5 mg mL^−1^, while 50% of survival was observed at 48 h. No toxicity was assessed at the concentration of 2.5 mg mL^−1^, being therefore suitable for therapeutic applications. The antifungal effects of PLA at concentration of 2.5 mg mL^−1^ determined in vitro were expanded to an in vivo experiment in *G. mellonella*. All uninfected larvae or treated with PBS survived for three days. As shown in Figure 8, the fungal infection control larval group had a survival rate between 50−10% up to 72 h, whereas the PLA-treated group, in general, increased the survival rate to about 50%. Particularly, pre-treatment increased the survival at 72 h to 80% for ATCC90028 strain and C14, 70% for C7 and C19, and 50% for C17. In PLA post-treated larvae previously infected with ATCC90028 and C7, survival increased up to 50%; when infected with C14 and C19, survival increased up to and 60 and 40%, respectively, and even up to 90% when infected with C17. From these results, it is possible to speculate that the treatment compound provided varying degrees of protection to the larvae, implying that the compound has different modes of action.

## 4. Discussion

VVC represent the most prevalent fungal infections that affects 5–8% of women of reproductive age with a considerable negative impact on the quality of their social and sexual lives. It is a difficult-to-manage infection because the complexity of its physiopathology leads to a gradual increase in drug resistance toward common drugs, such as nystatin, miconazole, and fluconazole, the latter being the first-line treatment for VVC [33]. Moreover, the intrinsic biofilm formation ability of *Candida* spp. has been recognized as one of the major critical factors for the antimicrobial resistance able to cause harmful recurrent chronic fungal infections. The antimicrobial activity of natural products which are effective against several microbial organisms responsible for common infectious and has little or no side effect has encouraged the development of treatments alternative to conventional antifungals [34,35]. Recently, PLA has received considerable interest as an antimicrobial agent. PLA is a natural organic acid produced by some lactic acid bacteria strains via lactate dehydrogenase and has been demonstrated to possess antimicrobial activity against both Gram-positive and Gram-negative bacteria, as well as against fungi. It has been demonstrated to possess capacity to prolong food shelf life due to a good hydrophilicity that makes it easily diffuse into the food matrix; in addition, it is odorless, unlike other organic acids, such as acetic acid and lactic acid [36].

In the present study, we first have selected *C. albicans* from patients with suspected VVC, and between these, we have chosen the four isolates that showed FLC resistance. Next, we demonstrated the antifungal activity of PLA. The MIC value for PLA against the reference strain of *C. albicans* ATCC90028 and the four clinical isolate was 7.5 mg mL^−1^ and MFC value was 7.5–10 mg mL^−1^. These values were similar to that determined for other fungi in previous studies, which repot antifungal activities in the range 6.5–12.0 mg mL^−1^ [37]. Moreover, candida cells were significantly inhibited when exposed to PLA in a time-dependent and dose-dependent manner as showed by the time to kill test. These results together confirmed the fungicidal activity of PLA.

Biofilm, an organized cell community, is a major virulence characteristic of *C. albicans* and the leading reason for ineffective treatment and infection recurrence. Another aim of this study was to determine the antifungal effect of PLA on biofilm formation. X. Jiang et al. [38] reported that PLA at sub-inhibitory concentrations could inhibit biofilm formation of *L. monocytogenes*, while higher concentrations of PLA (1/2MIC, 1 × MIC and 2 × MIC) reduced biomass and surviving cells in mature biofilms. J. Li et al. reported that the minimum concentrations of 3-phenyllactic acid, which significantly reduced dual species *S. mutans* and *C. albicans* biofilm formation, were 50 μg mL^−1^ [39]. In our context, all the four clinical isolates and the reference-strain were able to form biofilms ranging from moderate to strong ability. Additionally, these biofilms were inhibited by PLA at sub-MIC concentrations ranging from 0.6 to 5 mg mL^−1^. Additionally, 24 h of treatment with 2.5 mg mL^−1^ of PLA on preformed biofilms resulted in biofilm eradication ranging from 30 to 70%, as also shown by CLSM on a selected *C. albicans* mature biofilm. Our results show that the inhibition of biofilm formation and the breakdown of the preformed biofilm of *C. albicans* clinical isolates by PLA were significantly high, indicating that the compound exerts a good prospect for the treatment of this biofilm-related infection. Therefore, to understand how PLA could interfere with the molecular mechanisms of biofilm formation, we investigated changes in the level of genes expression of *ALS3*, *HWP1*, and *ERG11* considered to be some of the regulator factors in controlling biofilm formation [40]. Remarkably, after treatment with 2.5 mg mL^−1^ PLA, the adhesion gene, *ALS3*, was significantly down-regulated in the four *C. albicans* isolates (C7, C14, C17, and C19) and the reference strain. This is important because in a fungal biofilm context, these adhesins maintain microbial community cohesion by making cell–cell connections. Filamentous growth of *C. albicans* switching from yeast to hyphae is considered a critical virulence factor strictly connected to biofilm formation because this process support the development of stable structures [41]. In the present work, we observed up-regulation of *HWP1*, which encodes a fungal cell wall protein required for hyphal development and directly associated with biofilm formation, during PLA exposure in all strains except C17. It is worth noting that the PLA concentration used was only 2.5 mg mL ^−1^, a value that was not able to induce germination of the strains examined, differently from higher concentrations which were able to induce germ tube formation [42,43].

Treatment with PLA has up-regulated expression of *ERG11* in *C. albicans* clinical isolates C7 and C14, showing an undesirable trait of potential antifungals, also considering the reduced susceptibility to FLC as the *ERG11* gene encodes the azole target, CYP51, and the application of any compound that might increase *ERG11* expression could lead to resistance. However, the down-regulation in all other strains tested suggested that this compound has potential for further development as part of novel antifungal strategies. Although our study did not evaluate the comprehensive mechanisms associated with biofilm inhibition activity, suppression of *C. albicans* adhesion implies that PLA can prevent biofilm. The larvae of *G. mellonella* a useful pre-clinical in vivo test are simple, inexpensive, and quick screening model to evaluate microbial pathogenicity as well as host–pathogen interactions and its response shares similarities with the mammalian innate immune system [44]. The in vivo data confirmed the non-toxicity of the green compound PLA and reinforced the antifungal action as potential antimicrobial candidate against pathogenic *C. albicans*. In our experiments, the survival rate was markedly increased by PLA treatment both when used before and after infections.

In conclusion, we highlight the significant antifungal effect of PLA against clinical *C. albicans* isolates from patients with vulvovaginal candidiasis that were less sensitive to the conventional FLC used.

## Figures and Tables

**Figure 1 jof-09-00355-f001:**
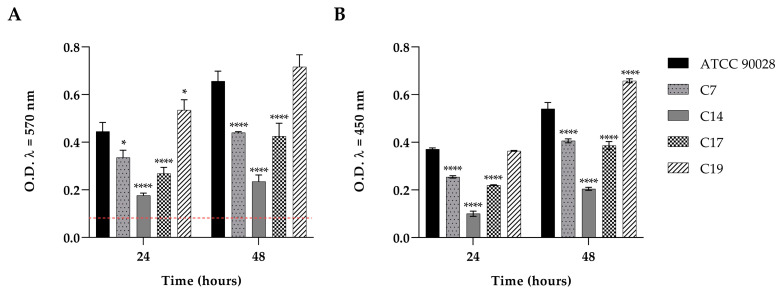
Panel (**A**): Comparison between the total biomass of biofilms of *C. albicans* ATCC90028 and *C. albicans* clinical isolates (C7, C14, C17, and C19) at 24 and 48 h. Red line represents ODcut (ODcut = mean of negative control with addition of 3 times). Panel (**B**): Comparison between the metabolic activities of the same strains at 24 and 48 h. Statistical significance: * *p* < 0.05, **** *p* < 0.0001 (Tukey’s test).

**Figure 2 jof-09-00355-f002:**
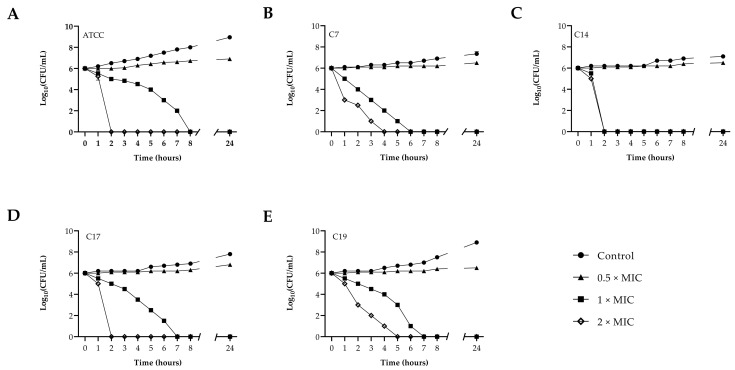
Time-kill kinetics of PLA against *C. albicans* ATCC90028 (panel **A**) and fluconazole-resistant *C. albicans* clinical isolates (panel **B**–**E**).

**Figure 3 jof-09-00355-f003:**
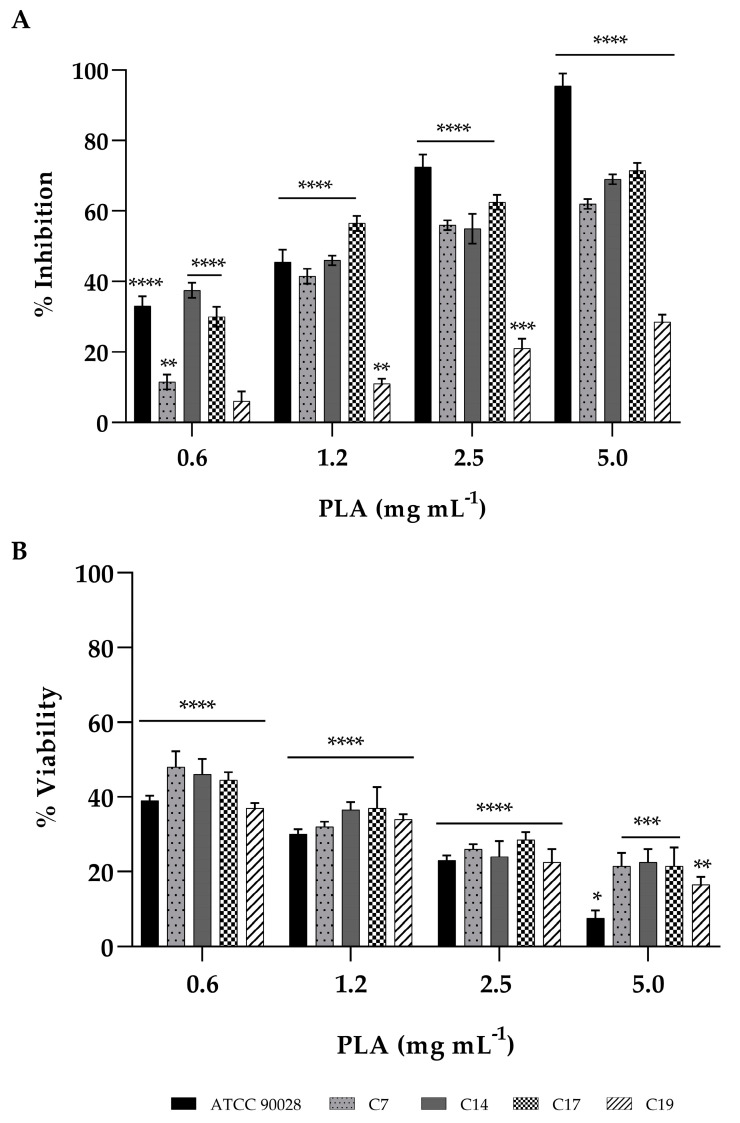
Inhibition of biofilm formation by PLA at concentrations of 0.6, 1.2, 2.5, and 5 mg mL^−1^ against *C. albicans* ATCC90028 and clinical isolates C7, C14, C17, and C19. Two different methodologies were used to quantify biofilm: crystal violet assay (panel **A**), which measures biofilm total biomass, and XTT assay (panel **B**), which measures metabolic activity. Data represent the mean (± standard deviation, SD) of three independent experiments, and each one was carried out with triplicate determinations. For all experimental points, * *p* < 0.05, ** *p* < 0.01, *** *p* < 0.001, or **** *p* < 0.0001 were obtained for treated samples versus untreated samples (Tukey’s test).

**Figure 4 jof-09-00355-f004:**
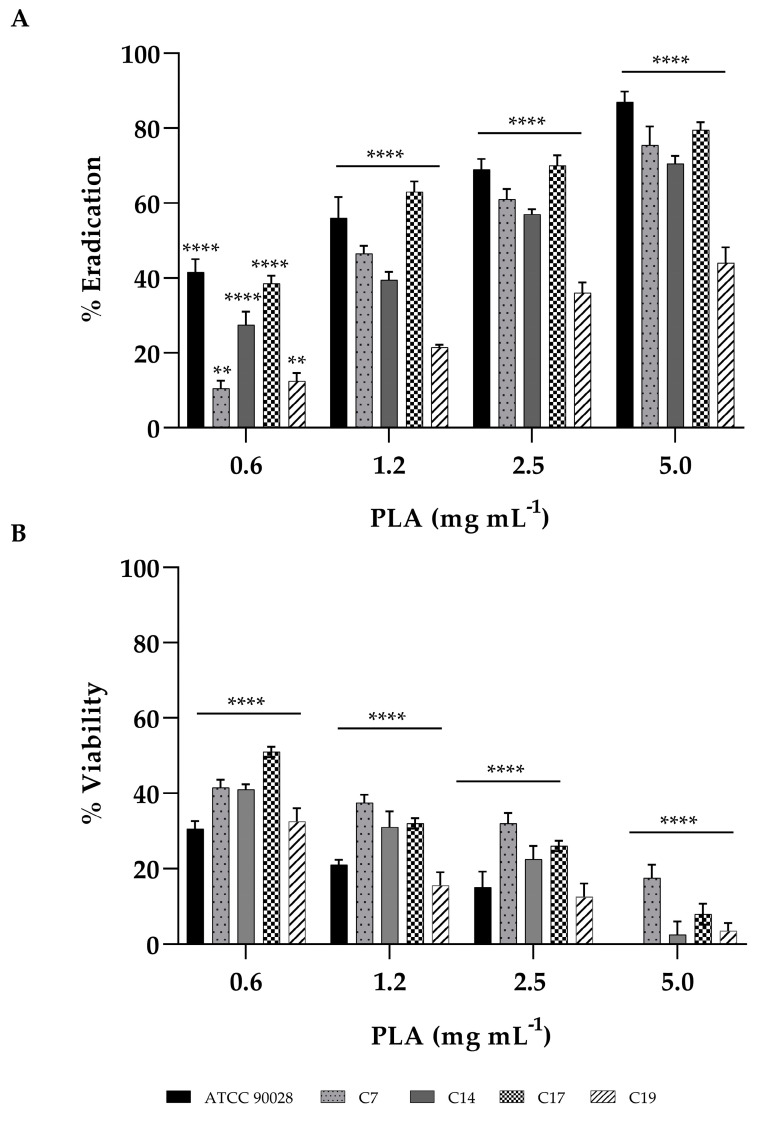
Eradication of pre-formed biofilm by PLA at concentrations of 0.6, 1.25, 2.5, and 5 mg mL^−1^ against *C. albicans* ATCC90028 and clinical isolates C7, C14, C17, and C19. Two different methodologies were used to quantify biofilm: crystal violet assay (panel **A**), which measures biofilm total biomass, and XTT assay (panel **B**), which measures metabolic activity. Data represent the mean (±standard deviation, SD) of three independent experiments, and each one was carried out with triplicate determinations. For all experimental points, ** *p* < 0.01, or **** *p* < 0.0001 were obtained for treated samples versus untreated samples (Tukey’s test).

**Figure 5 jof-09-00355-f005:**
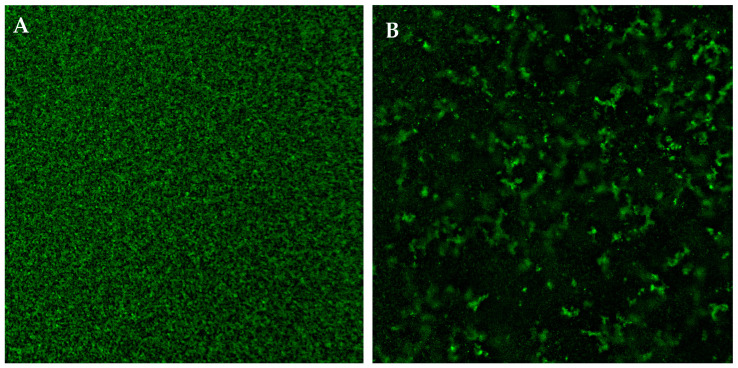
Effect of PLA on biofilm morphology. *C. albicans* 90028 biofilm formed for 48 h in the absence of PLA (**A**) and treated with 2.5 mg mL 1-PLA (**B**). The morphology of biofilms was visualized using a Zeiss LSM700 confocal microscope at 10× magnification.

**Figure 6 jof-09-00355-f006:**
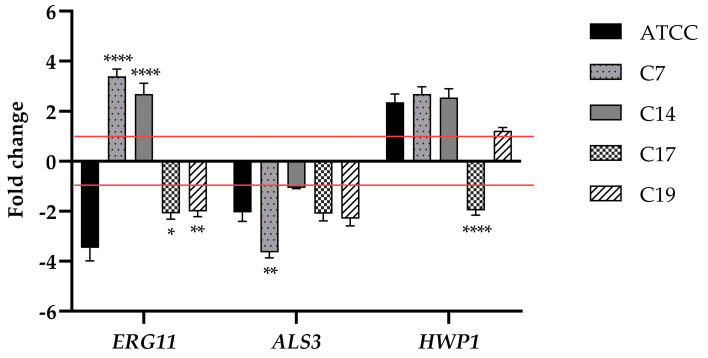
Expression analysis of selected genes of *Candida albicans* using real-time qPCR (*ERG11, ALS3*, and *HWP1*) in response to PLA. Histograms represent the fold differences in the expression levels of the genes selected during inhibition of biofilm with PLA at concentration of 2.5 mg mL^−1^. Red lines show fold change thresholds of −1 and +1, respectively. * *p* < 0.05, ** *p* < 0.01, and **** *p* < 0.0001 vs. *C. albicans* (Tukey’s test).

**Figure 7 jof-09-00355-f007:**
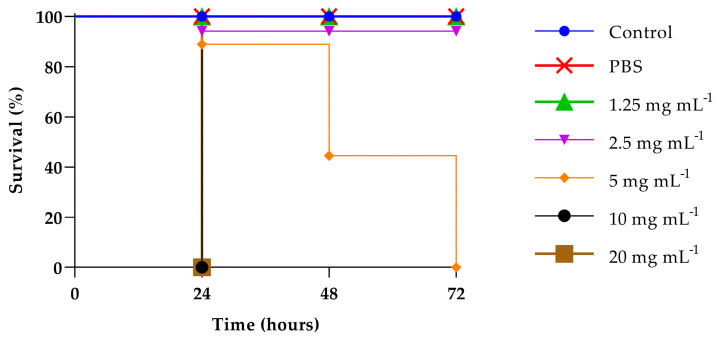
PLA toxicity on *G. mellonella* larvae treated with concentrations of 1.25 mg mL^−1^, 2.5 mg mL^−1^, 5 mg mL^−1^, 10 mg mL^−1^, and 20 mg mL^−1^.

**Figure 8 jof-09-00355-f008:**
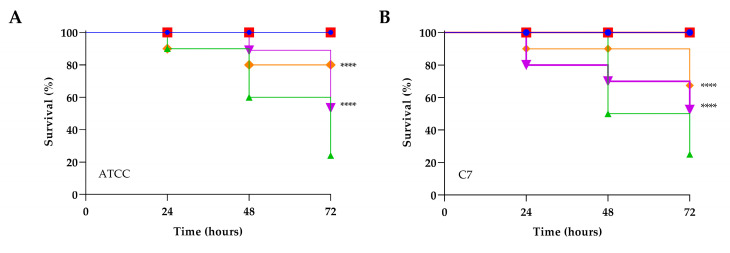
Kaplan–Meier plots of survival curves of *G. mellonella* larvae infected with *C. albicans* ATCC90028 (**A**), C7 (**B**), C14 (**C**), C17 (**D**), and C19 (**E**). The concentration of microorganisms was 1 × 10^6^ CFU/larva. All groups were treated with 2.5 mg mL^−1^ PLA before or after infection. All groups were compared with control (infected larvae). In all panels, survival curves of intact larvae and larvae injected with PBS are reported. **** represents *p*-value < 0.001 (Tukey’s).

**Table 2 jof-09-00355-t002:** Molecular characterization, following Sanger sequencing, of *Candida* spp. strains isolated from vaginal specimens.

Simple	Microorganism	Accession n.
C1	*C. tropicalis* 16	EF216862.1
C2	*C. albicans* SKH402	MT640023.1
C3	*C. albicans* SKH402	MT640023.1
C4	*C. glabrata AO34*	KX273821.1
C5	*C. albicans* SKH402	MT640023.1
C6	*C. albicans* SKH402	MT640023.1
C7	*C. albicans* AUMC13495	MH534919.1
C8	*C. haemulonii* LIP Ch9	KJ476201
C9	*C. albicans* SKH402	MT640023.1
C10	*C. dubliniensis* 94234	KF537273
C11	*C. albicans* AUMC13532	MH534924.1
C12	*C. albicans* JRP61	MF276778.1
C13	*C. albicans N96C*	KP675681.1
C14	*C. albicans* n43b	KP675647.1
C15	*C. glabrata RCPF 1398*	MK998697.1
C16	*C. dubliniensis* 95677	KF537274
C17	*C. albicans* n99b	KP675684.1
C18	*C. albicans P51L1*	KP878253.1
C19	*C. albicans* 123D	KP765027.1
C20	*C. albicans* AUMC13532	MH534924.1
C21	*C. albicans* JRP61	MF276778.1
C22	*C. dubliniensis* 98277	KF537275
C23	*C. albicans* SKH402	MT640023.1
C24	*C. albicans* m62a	KP675249.1
C25	*C. orthopsilosis* 1719	JQ023169
C26	*C. albicans FC6841*	MH628212.1
C27	*C. albicans SKH402*	MT640023.1
C28	*C. albicans N96C*	KP675681.1
C29	*C. haemulonii* LIP Ch4	KJ476196
C30	*C. albicans P51L1*	KP878253.1
C31	*C. albicans FC6841*	MH628212.1

**Table 3 jof-09-00355-t003:** Antifungal susceptibility of *Candida albicans* isolated from vulvovaginal candidiasis. AMB = amphotericin B; FLC = fluconazole; CSF = caspofungin; S = sensitive; I = intermediate; R = resistant.

Semple	AMB	FLC	CSF
C2 *C. albicans*	S	S	S
C3 *C. albicans*	S	S	S
C5 *C. albicans*	S	S	S
C6 *C. albicans*	S	S	S
C7 *C. albicans*	S	R	S
C9 *C. albicans*	S	S	S
C11 *C. albicans*	S	S	S
C12 *C. albicans*	S	S	S
C13 *C. albicans*	S	S	S
C14 *C. albicans*	S	R	I
C17 *C. albicans*	S	R	S
C18 *C. albicans*	S	S	S
C19 *C. albicans*	S	R	R
C20 *C. albicans*	S	S	S
C21 *C. albicans*	S	S	S
C23 *C. albicans*	S	S	S
C24 *C. albicans*	S	I	S
C26 *C. albicans*	S	S	S
C27 *C. albicans*	S	S	S
C28 *C. albicans*	S	S	S
C30 *C. albicans*	S	S	S
C31 *C. albicans*	S	S	S

**Table 4 jof-09-00355-t004:** Antifungal activity of PLA against *Candida albicans* isolated from vaginal secretion samples, expressed as minimal inhibitory concentration (MIC; mg mL^−1^) and minimum fungicidal concentration (MFC).

	PLA (mg mL^−1^)			
	MIC	MFC	MFC/MIC Ratio	
ATCC90028	7.5	7.5	1.0	fungicidal
C7	7.5	7.5	1.0	fungicidal
C14	7.5	7.5	1.0	fungicidal
C17	7.5	10.0	1.3	fungicidal
C19	7.5	10.0	1.3	fungicidal

## Data Availability

Not applicable.

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
