# Peer review of "Anti-Biofilm Activity of Phenyllactic Acid against Clinical Isolates of Fluconazole-Resistant Candida albicans"

_jof, 2023, doi:10.3390/jof9030355_

Round 1

Reviewer 1 Report

In this manuscript Maione et al., describes the antibiofilm activities of Phenyllactic acid against clinical and QC strains of C. albicans. Over all this is a well constructed study, however I suggest the authors to address the following comments. 

1) The observed MIC values of phenyllactic acid is significantly higher than the existing drugs. It is not clear, how a compound with poor potency will benefit in the clinics. 

2) For the MIC assay the authors used TSB as media, CLSI recommends RPMI-media for the antifungal studies. The authors have to follow CLSI recommendations to determine the MIC and MFCs. 

3) Results shown in the Fig indicate that clinical isolates except C19 do not form Biofilm as efficiently as ATCC strain. Interestingly, results shown in the Fig 2 indicates that only ATCC and C19 strains grow efficiently in the media used. In the cases of other clinical isolates, not even a single log increase in the CFUs are observed even after 24 h, suggesting TSB may not be the ideal media. Did the authors use any different media such as commonly used YPD broth to evaluate the growth. The authors required to use a media where all the isolates show comparable growth. The observed reduction in the biofilms could be simply due to the fact that strains do not grow the in the media used. 

4) The authors required to provide the seeding density used for the Biofilm formation assays. It is not clear what the authors meant by standard inoculum, if they have used O/N cultures, were the cell densities were same across all isolates? 

5) The authors are recommended to include standard test compounds with known MICs break points against C. albicans along with the AST assays conducted with phenyllactic acid.

6) The authors have given the CSF as abbreviation for  caspofungin, but there are instances in the manuscript where the authors have used CSA for the same. The authors need to correct this and use it consistently through out the manuscript.  

Author Response

In this manuscript Maione et al., describes the antibiofilm activities of Phenyllactic acid against clinical and QC strains of C. albicans. Over all this is a well constructed study, however I suggest the authors to address the following comments.

1) The observed MIC values of phenyllactic acid is significantly higher than the existing drugs. It is not clear, how a compound with poor potency will benefit in the clinics.

R: We thank the referee for his/her comments. Indeed, PLA has MIC values higher than the existing conventional drugs, among which the most commonly used fluconazole (FLC). However, due to the acquisition of resistance against FLC, in recent years the search for innovative alternative drugs, possibly green compounds, has been greatly explored. In this context, we have proposed PLA as a possible antifungal against FLC resistant Candida strains of clinical interest. It could be used for the topical treatment of Candida infections, also associated to biofilm formation, due to its low toxicity.

2) For the MIC assay the authors used TSB as media, CLSI recommends RPMI-media for the antifungal studies. The authors have to follow CLSI recommendations to determine the MIC and MFCs.

Thanks for the comment. The TSB was used only for pre-cultures, for the MIC assay was used RPMI. We have corrected in the text.

3) Results shown in the Fig indicate that clinical isolates except C19 do not form Biofilm as efficiently as ATCC strain. Interestingly, results shown in the Fig 2 indicates that only ATCC and C19 strains grow efficiently in the media used. In the cases of other clinical isolates, not even a single log increase in the CFUs are observed even after 24 h, suggesting TSB may not be the ideal media. Did the authors use any different media such as commonly used YPD broth to evaluate the growth. The authors required to use a media where all the isolates show comparable growth. The observed reduction in the biofilms could be simply due to the fact that strains do not grow the in the media used.

Thanks for the comment. As regards the Fig. 1, all strains can form biofilm even if with different capability. See lines 266-270. As showed in Fig. 2, all strains inoculated in RPMI media following the same condition of MIC assay, had an evident growth that decreased in the treated experiments. Lines 290-293

4) The authors required to provide the seeding density used for the Biofilm formation assays. It is not clear what the authors meant by standard inoculum, if they have used O/N cultures, were the cell densities were same across all isolates?

R: Cellular density was added to the manuscript.

5) The authors are recommended to include standard test compounds with known MICs break points against C. albicans along with the AST assays conducted with phenyllactic acid.

R: In Table 3 the antifungal susceptibility of standard test compounds (AMB, CSF, and FLC) is indicated because our aim was to identify Candida FLC resistant strains.

6) The authors have given the CSF as abbreviation for  caspofungin, but there are instances in the manuscript where the authors have used CSA for the same. The authors need to correct this and use it consistently through out the manuscript.

R: According to the reviewer comment, we modified as suggested.

Reviewer 2 Report

Dear Authors and Editor

The MS indicates the use of PLA as antifungal activity, however, authors need to revise the statistical data and indicate why PLA has activity against some isolates and did not show activity against other isolates either as phenotypes or RT-PCR

Introduction

-Please, indicate the previously reported antifungal activity of PLA

Method

-Section 2.6.

Detailed CV method need to be added and need to be represented as two separate parts ; inhibition of biofilm formation and inhibition of mature biofilm with more details how each of them was performed.

-In biofilm treatment, why authors used conc ranging from 0.6 to 5 mg mL, and why used conc 2.5 mg in RT-PCR assay

-RT-PCT lines 178-179 need rephrase

-Table 1; indicate the gene size, annealing temp and ref of the primers or designed in this study

- Section 2.9 need to be present before section 2.8 as the phenotypic characters are detected prior genotypic confirmation. Also, in the result section.

-Add ref to section 2.9

- Why authors just performed antimicrobial sensitivity to C. albicans and did not made for the other species?

- More details of the time kill study need to be added what the start bacterial count how it was attained? 

Results

Figure 2 line of the writing need to be clearer.

Figure 3 A; % Inhibition NOT %inhibition and the same for panel B

Statistical analysis is misleading as the * of significance in Fig 3 and 4 NOT indicating decrease in biofilm. How did you performed the ANOVA did you compared each treated strain with the untreated same strain NOT to the ST strain. 

Discussion

Please indicate why PLA has activity against some isolates and did not show activity against other isolates either phenotypes or RT-PCR

Minor

Line 56 : food industry NOT food Industry

Line 61:   its anti-biofilm NOT itsanti-biofilm

Line 83; Fifty patients NOT 50 patients

Line 224: Molecular analysis NOT Molecular Analysis

Author Response

Dear Authors and Editor

The MS indicates the use of PLA as antifungal activity, however, authors need to revise the statistical data and indicate why PLA has activity against some isolates and did not show activity against other isolates either as phenotypes or RT-PCR

Introduction

-Please, indicate the previously reported antifungal activity of PLA

We thank the referee for his/her comments. As reported in discussion (432-439), the MIC value for PLA against fungi in previous studies (doi:10.1007/s11947-008-0127-1) of 6.5–12.0 mg mL-1 was in line with our results both for C. albicans 90028 and the four FLC-resistant clinical isolate (7,5mg mL-1).

Method

-Section 2.6.

-Detailed CV method need to be added and need to be represented as two separate parts; inhibition of biofilm formation and inhibition of mature biofilm with more details how each of them was performed.

R: We thank the referee for his/her comments. More details on the CV method was added to the text.

-In biofilm treatment, why authors used conc.ranging from 0.6 to 5 mg mL, and why used conc 2.5 mg in RT-PCR assay

R: We thank the referee for his/her comments. The 2.5 mg mL-1 concentration had an antibiofilm effect but was not fun gicidal. For this reason, it was chosen to evidence possible variations on biofilm-related gene expression.

-RT-PCT lines 178-179 need rephrase

R: We agree with the referee and indeed we modified as suggested.

-Table 1; indicate the gene size, annealing temp and ref of the primers or designed in this study

R: According to the reviewer comment, we modified as suggested.

- Section 2.9 need to be present before section 2.8 as the phenotypic characters are detected prior genotypic confirmation. Also, in the result section.

R: According to the reviewer comment, we modified as suggested.

-Add ref to section 2.9

R: According to the reviewer comment, we added ref [27] to the section.

- Why authors just performed antimicrobial sensitivity to C. albicans and did not made for the other species?

R:. The paper has been focused on Candida albicans and in particular on C. albicans FLC resistant strains, since the primary involvement of this species in human infections, especially in the vulvovaginal region

- More details of the time kill study need to be added what the start bacterial count how it was attained?

R: Thanks for the comment, we modified in the text.

Results

Figure 2 line of the writing need to be clearer.

R: According to the reviewer comment, we modified as suggested.

Figure 3 A; % Inhibition NOT %inhibition and the same for panel

R: According to the reviewer comment, we modified as suggested.

Statistical analysis is misleading as the * of significance in Fig 3 and 4 NOT indicating decrease in biofilm. How did you performed the ANOVA did you compared each treated strain with the untreated same strain NOT to the ST strain.

R: Thanks for the comment. Asterisks indicated significative difference vs C. albicans ATCC. We modified the text for better clarification . 

Discussion

Please indicate why PLA has activity against some isolates and did not show activity against other isolates either phenotypes or RT-PCR

Thanks for the comment. As reported in the results, all C. albicans strains (ATCC 90028 and FLC resistant clinical isolates FLC-resistant isolates)  had a concentration-dependent sensitivity towards PLA, which resulted always effective. As regards gene expression results, a significant downregulation of ALS3 was found in all strains, whereas more strain dependent variability was observed in the other cases.   

Minor

Line 56 : food industry NOT food Industry

R: According to the reviewer comment, we modified as suggested.

Line 61: its anti-biofilm NOT itsanti-biofilm

R: According to the reviewer comment, we modified as suggested.

Line 83; Fifty patients NOT 50 patients

R: According to the reviewer comment, we modified as suggested.

Line 224: Molecular analysis NOT Molecular Analysis

R: According to the reviewer comment, we modified as suggested.

Round 2

Reviewer 1 Report

The authors have tried to address most of my concerns, but there are still some concerns that needs to be addressed.  Therefore, I suggest the authors address the following point 

1) The CFU data given in the Fig 2 indicates that ATCC strain and C19 strain grow more efficiently in the media used with a 2 log increase in growth for untreated controls,  while other strains do not grow as efficiently as these strains. It is not clear the observed differences in the biofilm formation (Fig 1) among these strains are due the difference in the growth rate observed for these strains in this media used. Did the authors test any other growth media for biofilm formation assays? If not I suggest the authors to include biofilm formation assay and growth curve in a rich media such as YPD. 

2) MICs of Phelyllactic acid was determined using broth dilution method, whereas the MICs of standard compounds were tested using disc diffusion method. I  suggest the authors to include one of the known antifungal agent with known MIC break point in the broth dilution assay along with phenyllactic acid as control compound.  

Author Response

The CFU data given in the Fig 2 indicates that ATCC strain and C19 strain grow more efficiently in the media used with a 2 log increase in growth for untreated controls,  while other strains do not grow as efficiently as these strains.

Thank you very much for the right observation. Our clinical strains exhibit different growth kinetics, so that the time to kill assay with PLA was performed in comparison to each proper control.

It is not clear the observed differences in the biofilm formation (Fig 1) among these strains are due the difference in the growth rate observed for these strains in this media used.

The biofilm formation was made using the RPMI medium that is elective to evaluate fungal biofilm capacity. Figure 1 reports that all strains were able to form biofilms, independently from their  growth rate. Apparently, biofilm formation depends on their different adhesive ability and subsequent colonization of the surface.

Did the authors test any other growth media for biofilm formation assays? If not I suggest the authors to include biofilm formation assay and growth curve in a rich media such as YPD.

Thank you very much for the suggestion, it is widely reported that medium composition may largely affect biofilm formation, so that an investigation on biofilm formation in different culture conditions will be the object of our next study.

MICs of Phelyllactic acid was determined using broth dilution method, whereas the MICs of standard compounds were tested using disc diffusion method. I  suggest the authors to include one of the known antifungal agent with known MIC break point in the broth dilution assay along with phenyllactic acid as control compound.  

Thank you very much for the comment. In our work, the disc diffusion standard method was used to evaluate the susceptibility of the strains to three conventional antifungals, aiming to select the fluconazole resistant strains for the following experiments with PLA. Instead, the MIC test was done to assess susceptibility of strains to PLA so that we used the untreated strain as proper control .

Reviewer 2 Report

he MS idea and concept is good. The authors made efforts in the experimental work. However, the statistics are not correctly represented, especially comparing the treated or untreated to standard. The authors should compare the treated to each control untreated sample. Also, the authors need to illustrate why some isolates are affected by  Phenyllactic acid and some isolates do not. All over, we cannot retrieve the conclusion that  Phenyllactic acid has antibiofilm activity by these two major points.

 the primers design of the RT PCR

The size of amplicon of one primer more than 800 bp and we never use more than 300   pb amplicon size.

Author Response

The MS idea and concept is good. The authors made efforts in the experimental work. However, the statistics are not correctly represented, especially comparing the treated or untreated to standard. The authors should compare the treated to each control untreated sample.

We want to thank the Reviewer for his/her appreciation. According to the Reviewer’s suggestion, in the revised manuscript statistics have been corrected, by comparing the treated sample with each control untreated one (see Fig. 3 and 4, and relative legends)  

 Also, the authors need to illustrate why some isolates are affected by  Phenyllactic acid and some isolates do not. All over, we cannot retrieve the conclusion that Phenyllactic acid has antibiofilm activity by these two major points.

Indeed, all strains show susceptibility to PLA, albeit to different extent. This is clearly evident in Fig. 2 , which shows that all strains were affected by PLA even if with different intensity. Further, PLA antibiofilm acvity is proved by the results of both Fig. 3 and Fig. 4, where inhibition and eradication effect of PLA is shown.

 the primers design of the RT PCR

The size of amplicon of one primer more than 800 bp and we never use more than 300   pb amplicon size.

Thank you very much for the right observation. This is a typos, it has been corrected in the manuscript

Round 3

Reviewer 1 Report

 The authors have responded to my concerns.

Reviewer 2 Report

Dear editors

The present study showed the Anti-Biofilm Activity of Phenyllactic acid against Clinical Isolates of Fluconazole-Resistant Candida albicans

The authors made the required corrections